# Estimating the human mutation rate from autozygous segments reveals population differences in human mutational processes

Vagheesh M. Narasimhan [1], Raheleh Rahbari [1], Aylwyn Scally[2], Arthur Wuster[1,3], Dan Mason [4], Yali Xue[1], John Wright[4], Richard C. Trembath[5], Eamonn R. Maher[6,7], David A. van Heel [8], Adam Auton[9], Matthew E. Hurles[1], Chris Tyler-Smith[1] & Richard Durbin [1]

Heterozygous mutations within homozygous sequences descended from a recent common ancestor offer a way to ascertain de novo mutations across multiple generations. Using exome sequences from 3222 British-Pakistani individuals with high parental relatedness, we estimate a mutation rate of $1.45 \pm 0.05 \times 10^{-8}$ per base pair per generation in autosomal coding sequence, with a corresponding non-crossover gene conversion rate of $8.75 \pm 0.05 \times 10^{-6}$ per base pair per generation. This is at the lower end of exome mutation rates previously estimated in parent–offspring trios, suggesting that post-zygotic mutations contribute little to the human germ-line mutation rate. We find frequent recurrence of mutations at polymorphic CpG sites, and an increase in C to T mutations in a 5′ CCG 3′ to 5′ CTG 3′ context in the Pakistani population compared to Europeans, suggesting that mutational processes have evolved rapidly between human populations.

[1] Wellcome Trust Sanger Institute, Hinxton, Cambridge CB10 1SA, UK. [2] Department of Genetics, University of Cambridge, Cambridge CB2 3EH, UK. [3] Department of Human Genetics and Department of Bioinformatics and Computational Biology, Genentech Inc., South San Francisco, CA 94080, USA. [4] Bradford Institute for Health Research, Bradford Teaching Hospitals NHS Foundation Trust, Bradford BD9 6RJ, UK. [5] Division of Genetics and Molecular Medicine, Faculty of Life Sciences and Medicine, King's College, London SE1 1UL, UK. [6] Department of Medical Genetics, University of Cambridge, Cambridge CB2 0QQ, UK. [7] Cambridge NIHR Biomedical Research Centre, Cambridge CB2 0QQ, UK. [8] Blizard Institute, Barts and The London School of Medicine and Dentistry, Queen Mary University of London, London E1 2AT, UK. [9] Department of Genetics, Albert Einstein College of Medicine, Bronx, NY 10461, USA. Vagheesh M. Narasimhan and Raheleh Rahbari contributed equally to this work. Correspondence and requests for materials should be addressed to V.M.N. (email: vagheesh@mail.harvard.edu) or to R.D. (email: rd@sanger.ac.uk)

In recent years, several approaches have been taken to estimating the human mutation rate, yielding results that differ substantially. These approaches can be grouped into three main categories: direct observation of mutations in present day parent–offspring comparisons (the direct rate), calibrating genetic divergence against fossil evidence for a past separation time (the phylogenetic rate)[1], or, more recently, population-genetic approaches that effectively estimate the ratio of the mutation rate to the recombination rate[2, 3]. For a genome-wide average mutation rate, the direct approaches have consistently estimated a rate of $1-1.25 \times 10^{-8}$ per base pair (bp) per generation, significantly lower than phylogenetic estimates, which suggest around $\sim 2 \times 10^{-8}$ per bp per generation[1] or estimates from population-genetic methods which suggest $1.6-1.7 \times 10^{-8}$ per bp per generation. Measurements of the mutation rate in coding sequence, obtained via the direct method applied to exome sequences of trios, are widely scattered but typically higher than the genome-wide rate at around $1.25-2.1 \times 10^{-8}$ per bp per generation[4]; the increase over genome-wide rates is usually attributed to differences in base composition giving higher frequencies of CpG dinucleotides, which are more mutable.

Many explanations have been suggested for why these estimates differ from each other[4, 5]. Possible shortcomings include: (a) small sample sizes, both in terms of the number of individuals the estimate is obtained from as well as the number of true de novo mutations (DNMs) detected; (b) inaccurate characterization of the false negative (FN) or false positive (FP) rates, perhaps because of comparisons of sequencing data with different properties from different individuals; (c) consideration only of mutations occurring in a single generation, leading to incomplete ascertainment of post-zygotic mutations in parents or offspring[6]; (d) incomplete allowance for the correlation with paternal age; (e) the inclusion of diseased individuals who might have a higher rate of DNMs; or (f) failure to account for gene conversion events.

To address these shortcomings, and to obtain an estimate which, like population-genetic approaches, averages over multiple generations and many mutational events, we adopted an approach based on observing heterozygous genotypes within sequence intervals inherited identical-by-descent (IBD) from a recent common ancestor (autozygous segments). Here we use exome sequences from healthy individuals with closely related parents, typically with $\sim 5\%$ of their genome autozygous in long (>10 Mb) segments. Heterozygote sites within autozygous segments can arise from DNMs in the generations since the common ancestor, or from gene conversions in the same period that led to the transfer of existing variants into one or the other IBD lineage, or from sequencing errors. We estimate the contribution of all three of these sources. Essentially the same approach was used previously on a small scale in a study of five individuals from the Hutterite cohort, and gave a genome-wide mutation rate estimate of $1.1 \times 10^{-8}$ per bp per generation[7]. The Palamara et al.[3] population-genetic method takes a similar approach, but makes a statistical estimate of the number of generations back to the most recent common ancestor in haplotype matches across individuals. In this study, we also compare our multi-generational estimate from autozygous segments with other previous estimates to understand the contribution of post-zygotic mutations to the overall human mutation rate. Additionally, as our estimate is one of the few data sets of DNMs obtained in a non-European population we examine differences in context-specific mutational spectra between human populations.

## Results

**Data set of 3222 exomes of high parental relatedness**. We analyzed exome sequences obtained from DNA from whole blood and sequenced to mean depth 27× from 3222 individuals of British Pakistani ethnicity[8]. The mean maternal and paternal age of the sampled individuals was 27.6 and 30.3 years respectively, a little below UK averages of 29 and 32 years respectively. These individuals are from communities with frequent first, second, and third cousin marriages, in a clan or "Biraderi" structure[9]. This level of relatedness allows us to examine DNMs accumulated across 6–10 meioses (Fig. 1). We restricted our analysis to autosomal single-nucleotide substitutions with the same genotype call from both samtools[10] and Genome Analysis Toolkit (GATK)[11] when calling across all samples.

**Estimating the mutation rate from autozygous segments**. To calculate the mutation rate, we first obtained $L$, the total length of the genome in which we counted heterozygous mutations. Previous work on this data set[8] showed that the locations of

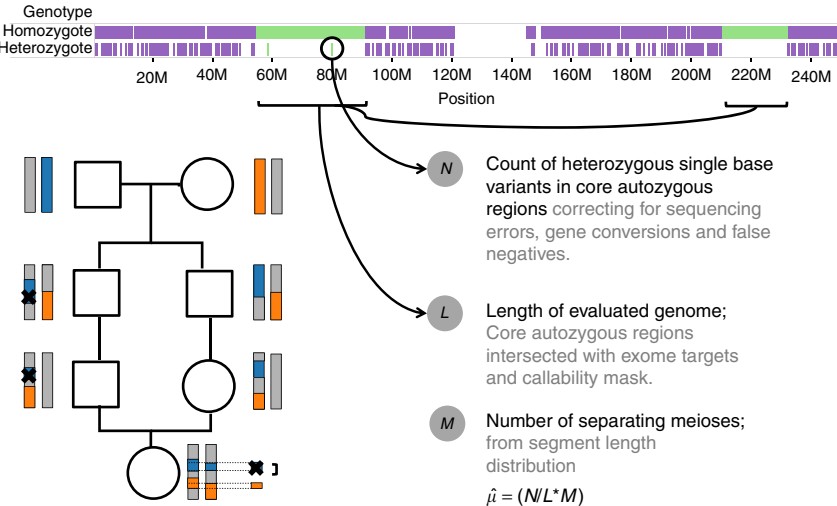

**Fig. 1** Study design: strategy to estimate the mutation rate. *Bottom left*: regions of the genome in an individual with first cousin parents are autozygous due to being inherited by two routes from a common founding chromosome. The X marks represent a DNM transmitted along the pedigree to the sequenced individual. *Top*: most sites in autozygous regions are homozygous, except for recent mutations, gene conversions and sequencing errors. *Bottom right*: the estimate $\hat{\mu}$ depends on three factors: *N, L* and *M*, as described in the text

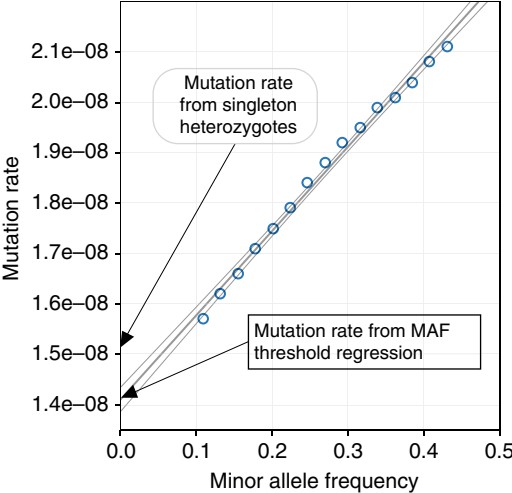

**Fig. 2** MAF-threshold regression to simultaneously obtain mutation rate and gene conversion rate. The mutation rate, is calculated by obtaining values of $N_f$ at different thresholds of minor allele frequency. The intercept on the y axis of the regression provides an estimate of the mutation rate that is corrected for gene conversion and the slope is used to calculate the estimate of the gene conversion rate

autozygous segments across individuals are randomly distributed with a mean of 210 individuals autozygous at each site. To enrich for segments that truly result from IBD we only consider segments that are at least 10 Mb long, as these arise in fewer than 8% of chromosome pairs that are separated by more than 10 meioses (Supplementary Fig. 1). To avoid calling mutations in segments adjacent to an autozygous stretch with a higher time to most recent common ancestor, we ignored the last 2 Mb at each end of the segment, having shown that truncating by more than this did not affect our estimate (Supplementary Fig. 2). We then took the intersection of the final set of autozygous core segments with the Illumina V5 exome bait regions and the 1000 Genomes Project accessibility mask[12] to yield a total evaluated length of $9.46 \times 10^9$ bp of DNA within the protein-coding regions of the genome.

Next, we estimated N, the number of heterozygous genotype calls within the autozygous sections, accounting for the FP and FN rates of the sequencing data. To estimate the FN rate, we simulated mutations by selecting a set of 10,000 random sites and switching the base in reads mapping there to an alternate base with probability 0.5. Then we remapped the modified reads, and measured the fraction of such simulated mutations that we could recall using our standard calling pipeline. To estimate the FP rate, we resequenced 176 individuals from whole blood taken at least 9 months apart using the same library preparation, sequencing protocol, and calling pipeline. We then modeled the replication rate of heterozygous mutations found in one sample and its duplicate, using a probabilistic framework that jointly accounts for both the FP and FN rates, as well as the allele frequency information of the site (Methods). For singletons (mutations seen just once in our samples) these approaches yielded a set of $N_0 = 1152$ heterozygous mutations with a FN rate of 17% and a FP rate of 1%. For mutations seen at allele frequencies above 10% (644 or more copies in 3222 samples) the estimated FN rate is lower, at 7.9%, since we used a multi-sample variant calling method (Methods, Supplementary Table 3).

Then, we determined M, the number of meioses leading to the most recent common ancestor, for each autozygous segment. We did this per individual, based on the autozygous segment length distribution in that individual. We used a supervised learning

approach that assigns the observed segment length distribution to an expected number of separating meioses, based on simulating recombinations in pedigrees with different degrees of relationship, according to the fine-scale recombination map. This yielded a weighted mean number of meioses across our entire data set of 6.63 (Methods). The inferred number of meioses per individual was in good agreement with the degree of relatedness from self-stated records for the approximately one third of our samples where this information was available (Supplementary Table 1).

**MAF-threshold regression to obtain the gene conversion rate.** Finally, we obtained mutation rate estimates in two different ways. First, we used the count of singleton heterozygotes $N_0$ to obtain the value $1.51 \times 10^{-8} \pm 0.05$ per bp per generation ($=N_0/LM$). Then we calculated a second value that was corrected for gene conversion by examining segregating variation in our data set. Here, we adopted an approach called minor allele frequency (MAF)-threshold regression[3], wherein we start from counts of $N_f$, the number of candidate heterozygous mutations in our truncated autozygous regions that have MAF less than $f$ in the whole cohort. For $f > 0$, $N_f$ will include alleles introduced by gene conversion, which occur at a rate proportional to the allele frequency. Therefore, we can use linear regression to obtain both the gene conversion rate (as the slope) and the mutation rate (as the intercept with the $f = 0$ axis). This approach yielded a single-nucleotide mutation rate of $1.41 \pm 0.04 \times 10^{-8}$ per bp per generation and a non-crossover gene conversion rate of $8.75 \pm 0.05 \times 10^{-6}$ per bp per generation (Fig. 2). This gene conversion rate estimate is a little higher than the previously reported rate of $6 \times 10^{-6}$ per bp per generation, which was obtained for whole genomes using phased trio data[13]. Our higher estimate for exome data may reflect higher recombination rates in coding sequence.

**Post-zygotic contribution to the human mutation rate.** The discrepancy between our two estimates for the mutation rate (1.51 and $1.41 \times 10^{-8}$ per bp per generation) is not statistically significant, but it is possible that our singleton estimate may be biased slightly upwards by including some gene conversions from rare alleles, whereas the regression estimate may be biased slightly downwards by removing some recurrent mutations. Thus we suggest a rounded summary estimate of $1.45 \times 10^{-8}$ per bp per generation. Overall, our estimates lie at the lower end of the published range for mutation rates in exome sequence, and below recent population-genetic estimates for the whole genome. More specifically, they are ~10% lower than the mean of the exome rates tabulated in Segurel et al.[4], with current generation mean paternal age also around 10% lower than that for the same set of studies. A concern for previous direct estimates based on a single generation is that post-zygotic mutations prior to separation of the germ line that lead to mosaicism could cause undercounting. However, our method covers the whole germ line life cycle in most of the generations, strongly mitigating such an effect if it exists. The fact that our estimates are not greater than previous exome estimates from trio studies, suggests that the contribution of post-zygotic mosaic-inducing mutations to the germ-line mutation rate is marginal[6, 14], in contrast with evidence from mouse[15] and cattle[16] that show that the earliest embryonic cell divisions account for ~25 and ~30% respectively of all mutations, although we acknowledge that this conclusion would be weakened if the unrecorded paternal age in previous generations for our cohort was considerably lower than that in the current generation.

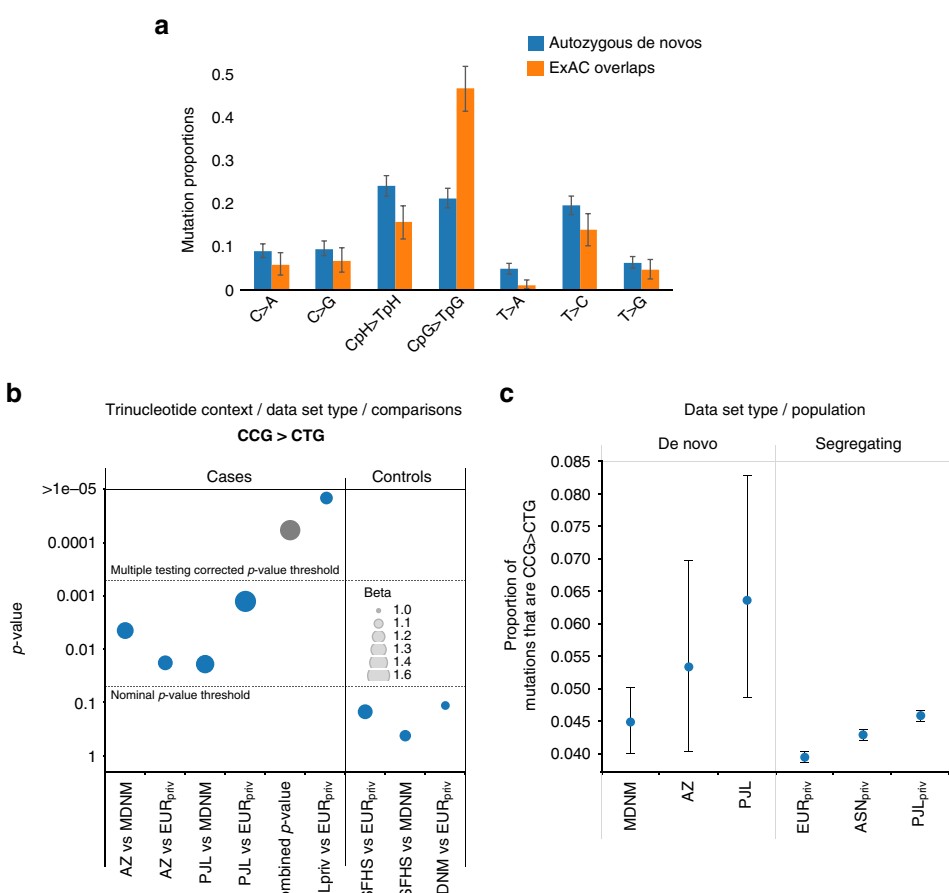

**Fig. 3** Signatures of DNMs and overlap of mutations with ExAC. **a** The distribution of de novo mutational signatures across all 1152 singleton candidate de novo mutations and 350 that overlap with ExAC. **b** Differences in context-specific mutation rate. *y*-axis: significance of the difference in proportion of 5′ CCG → CTG 3′ DNMs in 1152 mutations from the autozygosity data set (AZ) and 850 DNMs from the 1000 Genomes Project Complete Genomics trio data set (PJL) in comparison with 6948 mutations from the meta-analysis data set (*MDNM*) and variants private to Europeans in the 1000 Genomes Project (EURpriv). The combined *p*-value shows the result of meta-analysis of the AZ/MDNM and PJL/EURpriv comparisons. A comparison between private mutations in PJL and EUR in the 1000 Genomes Project population data set (PJLpriv) and (EURpriv), respectively is also shown. Significance of the difference in 747 DNMs from the Scottish Family Health Study (*SFHS*) is shown as a control; the size of the disk indicates the fold difference of the test as in the legend. **c** Proportion of mutations that are 5′ CCG → CTG 3′ for various whole genome data sets. The AZ data were adjusted for trinucleotide composition differences between exomes and whole genomes. PJLpriv variants show this increased context-specific mutation rate compared to both East Asians (ASNpriv) and Europeans (EURpriv) indicating that the increase is specific to Pakistanis

**Recurrence of a significant fraction of DNMs**. Comparing our DNMs to segregating variation seen in over 60,000 individuals from the Exome Aggregation Consortium (ExAC)[17], we found evidence for large-scale recurrence. Overall, 357 of 1152 (30.9%) of all our singleton DNMs were seen in ExAC, with a large proportion of these at CpG sites, the most mutable dinucleotide sites in the genome, for which ExAC is close to saturated[18] (Fig. 3a).

**Population differences in context specific mutation spectra**. Our ascertainment of DNMs is amongst the first in non-Europeans. Previous results that examined mutations private to each population from Phase 1 of the 1000 Genomes Project (1000GP) showed elevated rates of mutation in the tri-nucleotide context 5′ TCC 3′ → 5′ TTC 3′ in Europeans and to a lesser extent compared to Africans and to a lesser extent also in South Asians[19–21]. We therefore examined whether we could detect differences in mutational spectra between DNMs of South Asian and European ancestry (Supplementary Table 5). Here we compared the mutational spectra observed in our data set with

those from a meta-analysis of 6902 DNMs from whole-genome sequencing data of pedigrees of European ancestry[6]. After normalizing for the difference in sequence context between the datatypes, we found a difference in the proportion of a 5′ CCG 3′ → 5′CTG 3′ mutational signature that was nominally significant in our South Asian ancestry study compared to those from the European studies (ratio 1.35, $p = 0.0044$) (Fig. 3b). This replicated in a comparison of 850 genome-wide DNMs from a set of 15 trios from the Punjabi in Lahore, Pakistan (PJL) population from the 1000GP to the meta-analysis DNMs (ratio 1.42, $p = 0.019$). Both sets of Pakistani-ancestry DNMs were similarly significant when compared to a different control set of variants private to Europeans in the 1000 Genomes Project data (Fig. 3b), with a combined *p*-value for independent comparisons of $7.3 \times 10^{-5}$, which is experiment-wide significant across the 96 triplet mutation contexts.

As a second line of validation, we compared mutations private to the PJL population from the 1000 Genomes Project with the set of variants private to Europeans, which was again significant with *p*-value of $5.4 \times 10^{-37}$ (Fig. 3b). No other context showed such a consistent difference in effect or an experiment-wide significant

combined *p*-value, nor were there any experiment-wide significant differences for control comparisons using a set of 747 DNMs from the Scottish Family Health Study (SFHS)[6] (Supplementary Fig. 3). Also, comparisons with East Asian 1000 Genomes Project (ASNpriv) populations indicate that this difference is a relative increase specific to the PJL population, rather than a decrease specific to Europeans (Fig. 3c). We note that the signal from the comparison of private PJL to private EUR variants is present in all trinucleotide XpCpG contexts, and so would be consistent with a difference in the ratio of CpG to non-CpG mutation rates, perhaps due to changes in life history[22]. However, this is not the pattern that we see from the DNMs, where the consistent significant difference is restricted to the CCG context (GCG is also nominally significant in the autozygosity but not the PJL trio data, and ACG and TCG in neither).

## Discussion

We are not able to distinguish between possible biological causes for these differences, which might include environmental, life history or genetic factors such as population differences in methylation patterns or mismatch repair, or indeed any combination of these. However, we note that the fact that we see an effect in DNMs that contributes towards differences in population private mutations, which would be typically thousands of years old in the populations of 100 individuals used for this analysis, suggests that at least some factor or factors contributing to the Pakistani-specific mutation spectrum are ongoing in the UK Pakistani-ancestry community. The discovery of a second human sequence context with apparent differential mutation rates between continental populations supports and extends the observations by Harris[19] that mutational processes in at least some human populations have changed in the last 50,000 years, and is the first such effect to be seen in DNMs.

## Methods

**Cohort selection and variant calling.** We analyzed exome sequence data from a recent study of 3222 individuals of British Pakistani origin from Birmingham and Bradford. Full details of the sampling, sequencing and variant calling are available from the paper describing the data set[8], but we provide a brief overview here. These individuals were participants in either the UK Asian Diabetics Study[28] or the Born in Bradford study[23]. Ethical approvals were granted by the Bradford Research Ethics Committee; the Birmingham East, North and Solihull Research Ethics Committee; and the South Birmingham Research Ethics Committee and all individuals gave their consent to participate in the respective study. Individuals with severe long-term disease as reflected by their electronic health records and prescription rates were excluded. Exomes were sequenced in 75 bp paired end reads on the Illumina HiSeq platform from DNA from whole blood. Because that study was focused on identifying homozygous rare variants, the sequencing was at lower average coverage than standard for exome sequencing, with a mean coverage of 27× (per-individual and per-site coverage histograms are given in Supplementary Fig. 4). In addition, 176 samples with biological replicates collected at least 9 months apart were resequenced for quality control purposes using the same protocols.

Variant calling was performed by taking the intersection of two variant call-sets, one with GATK HaplotypeCaller[11] and one with samtools/bcftools[10]. Calling was restricted to the Agilent V5 exome bait regions +/− a 100 bp window on either end. The concordance between the two call-sets for SNPs was 95%. Discordant genotypes were set to missing and variant sites with > 1% missing genotypes were excluded. These calls were then run through a GATK VQSR training scheme at 99% True Positive Rate threshold using a set of single nucleotide polymorphisms (SNPs) from the phase 3 release of the 1000 Genomes Project cohort.

**Paternal age effect on mutation rate.** There is a known strong paternal age effect on mutation rate[18]. Our approach averages over several generations, and we were not able to obtain parental ages all the way back to the shared ancestor or the ratio of transmissions through the maternal and paternal germ lines. We obtained the average parental age at birth in this population by analyzing age information collected from the sampled individuals while they were admitted at a maternity ward during pregnancy. The mean maternal age in the present generation from this cohort was 27.6 years and the mean paternal age was 30.3, which are slightly lower than the average parental age in the UK overall, with mean paternal age of 32, and

maternal age of 29. Notably, our mean parental and maternal age estimates were within the range of the first direct estimate of the long-term generational interval estimated to be between 26 and 30 years[29]. Recent estimates of the mutation rate in coding sequence have been summarized in a recent review on the mutation rate[4]. The paternal age of our data set is at the lower end of those for exome studies, as is our corresponding overall mutation rate estimate (~10% lower than the mean in each case, though within the previous distribution).

**Estimating the FP and FN rate.** To obtain estimates of our FP sequencing error rate, we used 176 pairs of known duplicate samples that were sequenced and called with the same procedure and protocols and examined the dependence of the probability of replication of heterozygous calls $P(\text{het in dup 2}|\text{het in dup 1}, \alpha, \beta, f)$ on the FP rate $\alpha$, the FN rate $\beta$ and the allele frequency of the variant, $f$.

The replication rate, of seeing a heterozygote in duplicate 2, given that it is seen in duplicate 1, is given by equation 1 in Supplementary Note 1.

By the law of total probability, we can write this by conditioning on various scenarios of error and real genotypes (Supplementary Note 1, equation 2).

We then observed the replication rate empirically for each allele frequency from 0 to 1 in linear intervals of 0.01 to obtain an overconstrained system of 100 non-linear equations in $\alpha$ and $\beta$. To get an estimate averaged across all allele frequencies, we obtained solutions subject to the constraint that $0 < \alpha, \beta < 1$ and implemented this using the BBsolve package in R. Using this approach, we estimated a value for $\alpha$, 1%; and $\beta$, 9%.

In addition, we used an approach of introducing new artificial sequence variation on reads to obtain an independent estimate of the FN rate in our data. To do this, we picked 10,000 sites at random for which the reference allele was well defined (not reference N), and which were inside both the Illumina V5 exome baits and the 1000 Genomes Project callability mask, ensuring that selected sites were at least 100 bp away from each other (slightly longer than our read length). Then at each of these positions we decided on an alternate base to be synthetically introduced with 2/3 being transitions and 1/3 being transversions. Then, using a Bernoulli process ($p = 0.5$) for each read covering that site we switched the base of the selected position to the predetermined alternate base. The qualities, read lengths and insert sizes of these reads were maintained. We next removed the changed reads from the BAM and remapped them to the genome using the same command of BWA used to map the original data. We then proceeded to call variants at the given sites using the same calling procedure used to call the original data set (see above). Our estimate of FN rate is simply the number of introduced mutations that we failed recall using the above process.

As we performed joint calling across all 3222 exomes, variants seen in a single individual (i.e., singletons) were less likely to be called in comparison to shared variants with higher allele frequency. To adjust for this effect, we carried out the procedure of synthetically generating reads in multiple samples at various allele frequencies. In this setting, the FN rate was investigated in two ways. First, we calculated a rate for which we were unable to call the synthetically generated variable site in any sample. Second, we calculated a rate for which we were unable to call genotypes on an additional sample, given that the site was already known to be polymorphic. We report each of these categories of FN rates, along with their allele frequency (Supplementary Table 3). We find that there are significant differences in the FN rate between singleton mutations and those at higher allele frequencies. However, we find that there is little difference in our ability to call SNPs at frequencies above 10%, and use an average value of 7.9% FN rate for such frequencies.

**The length of evaluated genome in autozygous sections.** Using allele frequency information obtained from all 3222 individuals and the fine-scaled recombination map, we used BCFtools RoH[30] to obtain autozygous tract lengths as first reported in ref. [8]. These segments were found to be randomly distributed across the genome with any site autozygous in an average of 210 individuals.

To allow us to reliably infer the number of meioses giving rise to tract lengths, we chose to restrict ourselves to analyzing regions that could only arise from a very small number of recent generations, up to and including those from third cousins. To examine this, we used the R-package IBDsim[24] (see section on the predicted number of meioses from observed autozygous tract lengths) to simulate IBD sections in individuals separated by varying numbers of meioses. We then observed the longest autozygous block in each pedigree simulated 10,000 times, and found that fewer than 8% of pedigrees that are separated by more than 10 meioses have their longest autozygous segments longer than 10 Mb (Supplementary Fig. 1).

We then examined two further sources of bias that might affect the determination of the autozygous stretches. First, we might be overcalling regions because our Hidden Markov Model might be making an error by terminating a certain length after the end of a real stretch. This could introduce false heterozygous mutations and increase the estimated mutation rate. Secondly, segments that are IBD but separated by a larger number of meioses might lie directly adjacent to a long segment. These are more likely to have a higher number of heterozygous mutations per unit length, as mutations would have accumulated over more generations. To reduce the impact of both scenarios, we used an approach of truncating our regions by varying distances from each end and recalculating the mutation rate using only heterozygotes within the truncated sections. When we do this, there is no discernable change to the mutation rate

estimate beyond a truncation of 2 Mb (Supplementary Fig. 2). To ensure that the positions within these regions were themselves callable, we further restricted our evaluation to those that intersected the 1000 Genomes Callability mask, obtained from ftp://ftp.1000genomes.ebi.ac.uk/vol1/ftp/pilot_data/release/2010_03/pilot1/supporting/README_callability_masks. This resulted in a total length of callable genome of $9.46 \times 10^9$ bp of DNA.

**Predicting the number of meioses from autozygous tracts**. We infer the number of meioses separating the two chromosome pairs of the sequenced individual from the distribution of autozygous segment lengths. Chromosomes in the offspring of first cousins are separated by six meioses, and second cousins by eight meioses and so on. We began by simulating individuals who descend from pedigrees with varying parental relatedness from first cousin up to and including fourth cousin relationships. We simulated these recombinations in pedigrees using the R-package IBDsim[25], which uses the sex-specific fine-scale recombination maps, with random sex assignment through the pedigree. For each degree of parental relatedness, we simulated 10,000 pedigrees to obtain an empirical distribution of segment lengths and restricted our analysis to segments that are at least 10 Mb long. From these segment lengths obtained for each pedigree, we calculated three summary statistics that we used for inference: the length of the longest segment obtained, the average length of the segments and the total number of segments seen. Using these three features from the simulated data, we trained a supervised classification scheme to infer the number of separating meioses from a given segment length distribution. This was implemented using the supclust package in R that performs neighborhood component analysis for cluster assignment. As a validation of this approach, we compared our inferred parental relationships with those from self-stated relatedness and we report the most likely assignment for each individual along with information, if available, about their known self-stated relationship (Supplementary Table 1). As a second line of evidence, we obtained information on the segment length distribution obtained from well-characterized pedigrees where kinship was studied genetically from consanguineous families involved in rare disease studies[26]. In this evaluation, our approach inferred the pedigree relationships almost perfectly (Supplementary Table 2). Using the probabilistic assignment from our machine learning model of the number of meioses separating the chromosomes in individuals from our data set, and weighting this by the length of the genome that is autozygous in a particular individual, we calculated a weighted mean number of separating meioses across all the individuals of 6.63, i.e., between first and second cousin parental relatedness.

**Estimating the gene conversion rate**. Non-crossover gene conversion events require a copy of the alternate allele to be present on the chromosome from which the variant is copied, so can be modeled as occurring at a rate proportional to the allele frequency of the variant in the population. To obtain an estimate of the gene conversion rate, we utilized an approach known as maf-threshold regression[3]. To do this we computed the mutation rate using a range of maximum allele frequency thresholds, and performed a linear regression of the resulting mutation rate on the allele frequency threshold. The intercept of this regression on the y-axis (allele frequency 0) provides an estimate of the mutation rate that is corrected for gene conversion, while the slope corresponds to the gene conversion rate. We computed this regression line for allele frequencies between 10 and 50%. To obtain the mutation rate in this allele frequency range, we used the average FN rate across these frequencies of 7.9%, as obtained above. We also need to consider the population heterozygosity which determines the chance that a particular variant is present on a chromosome. The population heterozygosity in this data set is $9.56 \times 10^{-4}$ which is in line with other exome estimates from the 1000 Genomes Project. We computed standard errors for both the intercept and the slope by using a bootstrap procedure that we implemented using the boot package in R.

**Partitioning of DNMs into mutational spectra**. We subclassified the six distinguishable point mutations and their reverse complements (C:G → T:A, T:A → C:G, C:G → A:T, C:G → G:C, T:A → A:T and T:A → G:T) by calculating the relative frequency of mutations at the 96 triplets defined by the mutated base and its flanking base on either side[27]. For each of the trinucleotide classes, we compared the mutational signatures across sets of DNMs using a $2 \times 2$ table and tested whether the proportion of mutations of one class is significantly different in one population versus another. To be as conservative as possible, we used Yates continuity correction and corrected for multiple hypothesis due to the 96 tests we perform for each signature using the Bonferroni method. We show in Supplementary Table 4 the $2 \times 2$ table for one comparison of the 5′ CCG 3′ → 5′ CTG 3′ class of mutation that is discussed in the main text, and data for the counts for the samples published in this study are available in Supplementary Data 1 and the significance of the tests in Supplementary Fig. 3.

**Comparison of DNMs in the 1000 Genomes Project samples**. We defined derived SNPs that were private to each continent in the same manner as Harris 2015. For the African continent, however, we chose to differ slightly from the definitions used to define the 1000 Genomes Project phase 3 AFR category. We excluded populations from the Americas, which are known to have recent

admixture from both Africa and Europe, and so dropped ASW (African Americans from the Southwest US) and ACB (African Caribbeans from Barbados) from our African category. Therefore, we consider SNPs private to Africa if they are variable in at least one of the populations LWK (Luhya from Kenya), YRI (Yoruba from Nigeria), ESN (Esan from Nigeria), GWD (Gambian from western divisions of Gambia) and MSL (Mende in Sierra Leone) and are not variable in the South Asian, European and East Asian categories, as defined by the 1000 Genomes Project. Then we obtained SNPs that were private to each continental group with allele frequency at least two, to avoid any increased noise in singletons (as Harris 2015), and examined differences in their trinucleotide contexts as above for our set of DNMs.

**1000 genomes PJL trio DNMs discovery and validation**. Blood-derived DNA samples of 15 Punjabi trios from the Punjabi in Lahore, Pakistan (PJL) population of the 1000 Genomes Project were whole-genome sequenced by Complete Genomics (CG)[31], resulting in 12,496 candidate DNMs per trio on average. In our initial filtering, we removed calls seen in any other individual, or in the CG founder, and sites that were polymorphic in the 1000 Genome Project Phase 1. This resulted in 3609 candidate DNMs per trio. There were two criteria by which a putative DNMs were selected for validation: either they were genotyped as a de novo call using Samtools, or the de novo call had a quality score > 50 (i.e., ALT_EAF, as defined by CG). This resulted in 759 candidate DNMs per trio for validation. Candidate sites were validated by designing Agilent SureSelect probes for the candidate sites, followed by enrichment and sequencing on Illumina Hi-Seq. Overall, 850 sites were validated as DNMs (56.6 per trio on average).

Because of this ascertainment strategy, the validated trio de novos are suitable for assessing relative rates between different contexts, but not for measurement of absolute mutation rate. Notwithstanding this, the numbers that were validated within exome regions are consistent with the rates reported here, although the standard deviations of the estimates are large because the numbers are small. Of the 850 whole genome PJL trio DNMs, only 17 were in the accessibility-masked exome target region of ~ 45 Mb, and 32 were in the accessibility-masked call region with 100 bp extensions of ~ 82 Mb. Ignoring corrections for ascertainment, these would both give rate estimates of ~1.3 with standard deviation around 0.3 or 0.23 respectively, which is within two standard deviations of any of the current estimates.

**Data availability**. Data reported in the paper are available under a Data Access Agreement at the European Genotype-phenome Archive (www.ebi.ac.uk/ega) under accession numbers EGAS00001000462, EGAS00001000511, EGAS00001000567, EGAS00001000717 and EGAS00001001301[28–31].

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

## Acknowledgements

The study was funded by the Wellcome Trust (WT102627 & WT098051). This paper presents independent research funded by the National Institute for Health Research (NIHR) under its Collaboration for Applied Health Research and Care (CLAHRC) for Yorkshire and Humber. Core support for Born in Bradford is also provided by the Wellcome Trust (WT101597). Born in Bradford is only possible because of the enthusiasm and commitment of the Children and Parents in BiB. We are grateful to all the participants, health professionals and researchers who have made Born in Bradford happen. We would like to thank the Exome Aggregation Consortium and the groups that provided exome variant data for comparison. A full list of contributing groups can be found at http://exac.broadinstitute.org/about. Finally, we thank Anna Rutterford for useful discussions relating to the study design.

## Author contributions

V.M.N. conceived the mutation rate analysis. V.M.N., R.R., A.S., Y.X., C.T.-S. and R.D. interpreted the results; A.A. and A.W. performed the PJL trio de novo SNV discovery and validation; V.M.N. and R.R. performed the statistical and bioinformatic analyses. D.M., J.W., E.R.M., R.C.T., R.D. and D.A.v.H. designed, developed and managed the selection and sequencing of autozygous subjects, providing the data analyzed in this study. V.M.N. and R.D. drafted the manuscript. All authors contributed to the final version of the manuscript.

## Additional information

**Competing interests:** V.M.N. was supported by a Wellcome Trust Ph.D. Studentship (WT099769). E.R.M. is funded by NIHR Cambridge Biomedical Research Centre. R.D. and M.E.H. declare their interests as founders and non-executive directors of Congenica Ltd. R.D. also owns stock in Illumina Inc. from previous consulting and is a scientific advisory board member of Dovetail Inc. The remaining authors declare no competing financial interests.

