## [Peer Review File · Nature Communications]

Reviewers' comments:

Reviewer #1 (Remarks to the Author):

Nature Communications manuscript NCOMMS-16-16123-T is entitled: "A multi-generational estimate of the human mutation rate from autozygous segments reveals population differences in human mutational processes". The manuscript is submitted for consideration of publication by Narasimhan VM, Rahbari R, Scally A, Wuster A, Mason D, Xue Y, Wright J, Trembath RC, Maher ER, van Heel DA, Auton A, Hurler ME, Tyler-Smith C, Durbin R, principally of the Wellcome Trust Sanger Institute, Cambridge, England. The following brief summary is taken from the submitted manuscript.

The authors note that three main approaches have been taken as methods to estimate the human mutation rate. These broadly grouped are: 1) direct observation in parent-offspring comparisons ($1-1.25 \times 10^{-8}$ per bp per generation); 2) using fossil evidence or calibrating genetic divergence across time ($\sim 2.0 \times 10^{-8}$ per bp per generation); and 3) using populations to determine ratio of mutation rate to recombination rate ($1.6-1.7 \times 10^{-8}$ per bp per generation). While there have been many explanations proposed to explain these differences, the authors have taken the approach of "observing heterozygous genotypes within sequence intervals inherited identical-by-descent from a common ancestor". They report an analysis of exome sequences (mean coverage = 28x) from 3,222 individuals of British Pakistani ethnicity with a high degree of relatedness to estimate their de novo mutation rate (DNMs) and non-crossover gene conversion rate by utilizing the autozygous segments. The individuals in their study typically had $\sim 5\%$ of their genome autozygous in long (>10 Mb) segments. From their analysis using this data set with high parental relatedness, they estimate a "mutation rate of $1.45 \pm 0.5 \times 10^{-8}$ per base pair per generation in autosomal coding sequence with a corresponding non-crossover gene conversion rate of $8.75 \pm 0.05 \times 10^{-8}$ per base pair per generation." They propose, therefore, that post-zygotic mutations contribute little to the human germline mutation rate. The authors also observed in their data set frequent recurrence of mutations at polymorphic CpG sites. They also observed an increase in C to T mutations in a 5' CCG 3' \diamond 5' CTG 3' when comparing the Pakistani ancestry to Europeans and concluded that "mutational processes have evolved rapidly between human populations".

Comments:

1. The manuscript is well written, with a clear methods section. There are three figures in the main text. The supplementary section has three figures and five tables and is appropriately supportive of the main body of the manuscript.
2. The authors suggest that their estimated de novo mutation rate, which is at the low end of that reported from this analysis, is because post-zygotic mutations have little contribution to the germline mutation rate. Is it possible that this observation could have a different interpretation than simply the inclusion of post-zygotic mutations by prior analysis? There are also 15 P/JL trios that the authors obtained DNMs from, but did not provide an estimate of mutation rate in the exome region, which would be helpful to compare and quantify the proportion of high variant allele frequency post-zygotic mutations.
3. The authors showed that the rate of CCG \rightarrow CTG in the Pakistani population is significantly higher than that in European population. Is it possible that this is due to an elevated context-specific mutation rate in the Pakistani population compared to other populations; or because of a lower context-specific mutation rate in Europeans? The authors should attempt to answer this question, perhaps using the method similar to Harris' approach. Could there be a biological explanation of the signal observed?
4. Based on Supplementary Figure 3, P/JLPriv vs. EUR in GCG \rightarrow GTG, TCG \rightarrow TTG and ACG \rightarrow ATG are also significant after correction for multiple testing; and these are all CpG sites. Can the authors comment on these substitutions? Is it possible that the difference in population has to do with population specific methylation, or with the filtering process, where a mutation that has been seen in the European population is more likely to have been filtered?
5. On a related note, this recent manuscript by Mathieson and Reich also picked up some unique

signatures concentrated at CpG sites in Native American populations. I wonder if the authors can perform similar analysis in their cohort.

<http://biorxiv.org/content/early/2016/07/13/063578>

6. Some minor comments:

a. Second paragraph on Page 3, what is the FP rate for mutations with mutation rate >10%? Also 1%?

b. page 11, third paragraph, the following sentence seems to be incomplete: "As we are only interested in examining sections that are larger than 10Mb long, we only examined"

Reviewer #2 (Remarks to the Author):

Narasimhan et al. use exome sequencing and familiar relationships to estimate the mutation rate in autosomal coding regions in a very large cohort. The estimation of human mutation rate is an active area of interest and this work is a good addition to the field. The analyses presented are well thought-out, the results are interesting, and conclusions fit the data presented. The authors report a mutation rate on the lower end of what has been reported in previous studies, and I would suggest that the authors investigate this difference a little more. I have couple of suggestions for the authors:

1. The authors use as the evaluated genome the length of targeted sequencing in autozygous regions that intersect the 1000 Genomes Callability mask. Given the modest coverage of the sequencing data ~28X, I would suggest that the authors also take bases with sufficient coverage for variant calling into account. In addition, were the excluded/VQSR filtered variant sites subtracted from the evaluated genome? Since the mutation rate reported is at the lower end of exonic rates, it is important to show that this is not due to a evaluate genome length that is too large.
2. Related to point 1, the authors state that the mean coverage is ~28X. It would be informative to see the coverage distribution across individuals.
3. Another potential reason that the rate estimate is lower is that the paternal ages are on average lower across the generations analyzed relative to other estimates of the autosomal coding mutation rate. Although the authors only have information on the most recent generation, can they comment on how these ages compare to the paternal/parental ages in other publications of autosomal coding mutation rate.
4. In Supplementary Table 1, in the last line, there is an arrow where I think it should be >10.

We thank the reviewers for their helpful review and comments, we show these below (in italics) with our response (in normal text).

Response to Reviewer 1

The authors suggest that their estimated de novo mutation rate, which is at the low end of that reported from this analysis, is because post-zygotic mutations have little contribution to the germline mutation rate. Is it possible that this observation could have a different interpretation than simply the inclusion of post-zygotic mutations by prior analysis? There are also 15 P JL trios that the authors obtained DNMs from, but did not provide an estimate of mutation rate in the exome region, which would be helpful to compare and quantify the proportion of high variant allele frequency post-zygotic mutations.

The 1000 Genomes P JL trio de novo mutations were all validated, but our ascertainment strategy for them described in the supplementary text was (a) not exhaustive (it was not guaranteed to find them all, and we expect it missed some) and (b) was not done in such a way as to calculate the denominator. These issues don't affect our use of this de novo set to replicate the mutational spectrum results, which only depend on relative rates, but do mean that this data set is not calibrated for estimation of absolute mutation rate (indeed this is why an absolute mutation rate estimate was not published by the 1000 Genomes Project from these data). In any case, the numbers of observed de novos in exome regions are too small for an informative estimate to be obtained from these data: of the 849 whole genome P JL trio de novo mutations, only 17 were in the accessibility-masked exome target region of ~45Mb, and 32 were in the accessibility-masked call region with 100bp extensions of ~82 Mb. Ignoring further denominator corrections, these both give rate estimates of approximately 1.3 with standard deviation around 0.3 or 0.23 respectively, which is within two standard deviations of any of the current estimates. We now give these numbers in the supplementary text discussing the trios, with appropriate caveats about not being able to estimate the denominator accurately as well as being underpowered.

3. The authors showed that the rate of CCG>CTG in the Pakistani population is significantly higher than that in European population. Is it possible that this is due to an elevated context-specific mutation rate in the Pakistani population compared to other populations; or because of a lower context-specific mutation rate in Europeans? The authors should attempt to answer this question, perhaps using the method similar to Harris' approach. Could there be a biological explanation of the signal observed?

We have compared the de novo and private mutations in P JL (P JLpriv) now also to private mutations in the African and East Asian 1000 Genome Project populations as well as to Europeans, and report that these are increased in both P JL (p-values for P JLpriv vs Africans is $<10^{-16}$ and P JLpriv vs East Asians is 4.8×10^{-9}). If we compare the European (EUR) *de novo* and private CCG>CTG mutations to private mutations in Africa and East Asia we see that the Europeans have significantly higher rates than the Africans but significantly lower rates than East Asians (p-values for EUR vs Africans is 0 and for EUR v East Asia is 7.3×10^{-121}). We now discuss further in the manuscript possible biological causes for the signal, though because we don't have any information about the cause, such discussion is necessarily limited.

4. Based on Supplementary Figure 3, P JLPriv vs. EURPriv in GCG->GTG, TCG->TTG and ACG->ATG are also significant after correction for multiple testing; and these are all CpG sites. Can the authors comment on these substitutions? Is it possible that the difference in population has to do with population specific methylation, or with the filtering process, where a mutation that has been seen in the European population is more likely to have been filtered?

We now discuss more extensively the observation that all xCG mutations are enriched in the population private comparison, whereas only CCG consistently in the de novo mutations. We also refer to the

possibility of population differences in methylation as a cause, but are not able to distinguish this from other possible mechanisms.

5. On a related note, this recent manuscript by Mathieson and Reich also picked up some unique signatures concentrated at CpG sites in Native American populations. I wonder if the authors can perform similar analysis in their cohort. <http://biorxiv.org/content/early/2016/07/13/063578>

We do not report population genetic data from a set of diverse populations like those used by Mathieson and Reich. Our approach is complementary; i.e. to test for an effect in the present generation using datasets of de novo mutations.

<http://biorxiv.org/content/early/2016/07/13/063578>

6. Some minor comments:

a. Second paragraph on Page 3, what is the FP rate for mutations with mutation rate >10%? Also 1%?

We presume the author means with allele frequency >10%, not with mutation rate as written. As described in our methods section, the way we calculate FP rate relies on a set of measurements across a range of allele frequencies. We report the corresponding FP and FN rates across a range of allele frequencies below using this method. We have added these to supplementary table 3.

Allele Frequency	FP rate	FN rate
>10%	1.05%	8.24%
>20%	0.86%	8.04%
>30%	0.94%	8.19%

Our method does not allow us to calculate a rate just for the 1% bin. Also we don't use a 1% bin for any other estimates. If we look at sites seen just once in the 176 duplicates but not in any of the other samples, i.e. singletons, there are 557 that are replicated as heterozygous calls, and in 42 cases such a site is only called in one of the two replicates. Attributing all these 42 to false positive, and ignoring false negatives as a likely cause for some of these, this gives an upper bound on singleton false positives of $42/(577*2+42) = 3.6\%$, which is a little higher than our overall false positive estimate of 1%, but is biased upwards to an unknown extent by ignoring false negatives. We can't correct with our estimated false negative rate because we don't know the dependency for repeated false negatives at the same site, which matters for this calculation, but not for the correction of the denominator of the mutation rate estimate, where we are looking at the probability of missing a singleton once, not missing two mutations at the same site.

b. page 11, third paragraph, the following sentence seems to be incomplete: "As we are only interested in examining sections that are larger than 10Mb long, we only examined"

This was incorrectly left in the original manuscript during final editing, and has been removed.

Response to Reviewer 2:

Narasimhan et al. use exome sequencing and familiar relationships to estimate the mutation rate in autosomal coding regions in a very large cohort. The estimation of human mutation rate is an active area of interest and this work is a good addition to the field. The analyses presented are well thought-

out, the results are interesting, and conclusions fit the data presented. The authors report a mutation rate on the lower end of what has been reported in previous studies, and I would suggest that the authors investigate this difference a little more. I have couple of suggestions for the authors:

1. The authors use as the evaluated genome the length of targeted sequencing in autozygous regions that intersect the 1000 Genomes Callability mask. Given the modest coverage of the sequencing data ~28X, I would suggest that the authors also take bases with sufficient coverage for variant calling into account. In addition, were the excluded/VQSR filtered variant sites subtracted from the evaluated genome? Since the mutation rate reported is at the lower end of exonic rates, it is important to show that this is not due to a evaluate genome length that is too large.

We addressed this issue of the callable denominator in a different way than by setting depth thresholds. After applying the 1000 Genomes callability mask, we simulated the consequences of 10,000 hypothetical mutations in our data set and assessed our ability to call them by running through our entire read mapping and calling pipeline, showing that this reduced our power to call population singletons by ~16% and higher frequencies by ~8% (details in Supplementary Text and Supplementary Table 3, now extended), equivalently reducing the denominators for our point-estimate and MAF-regression estimates by these amounts respectively. This approach automatically takes into account variation in read depth and in mapping ability, and excludes sites filtered out by VQSR and other elements in the calling process. It is therefore more direct and less biased than to try to identify a priori the set of callable locations.

2. Related to point 1, the authors state that the mean coverage is ~28X. It would be informative to see the coverage distribution across individuals.

We now provide the average mapped coverage per-site and per-individual in the evaluated regions (exome target intersecting the 1000 Genomes Project callability mask) in two supplementary figures.

3. Another potential reason that the rate estimate is lower is that the paternal ages are on average lower across the generations analyzed relative to other estimates of the autosomal coding mutation rate. Although the authors only have information on the most recent generation, can they comment on how these ages compare to the paternal/parental ages in other publications of autosomal coding mutation rate.

We have compared the paternal/parental ages with those from the UK national averages, as well as a long term direct estimates of the average human generation time as measured through ancient DNA. We now also discuss comparisons of paternal age of the most recent generation in our study with those from other published datasets as compiled in a review by Segurel et al. 2014 (see figure below for the purposes of this review process, which superimposes our estimate of the mutation rate marked in a blue circle on their figure), and we explicitly state that if the paternal age in previous generations was even lower, then that could affect the strength of our conclusions.

4. In Supplementary Table 1, in the last line, there is an arrow where I think it should be >10 .

We have now corrected this to “ >10 (not considered)”.

REVIEWERS' COMMENTS:

Reviewer #1 (Remarks to the Author):

I have reviewed the authors' responses to the questions and comments of the initial reviews. They have adequately addressed each and made the appropriate changes to the revised manuscript. I recommend that the manuscript be accepted for publication.

Reviewer #2 (Remarks to the Author):

The authors have addressed my concerns.